# Everyday Object Meets Vision-and-Language Navigation Agent via Backdoor

Keji He[*1]   Kehan Chen[*2,3]   Jiawang Bai[†4]   Yan Huang[2,3]
Qi Wu[5]   Shu-Tao Xia[6]   Liang Wang[†2,3]

[1]Shandong University
[2]New Laboratory of Pattern Recognition
Institute of Automation, Chinese Academy of Sciences
[3]School of Artificial Intelligence, University of Chinese Academy of Sciences
[4]Tencent
[5]School of Computer Science, University of Adelaide
[6]Tsinghua Shenzhen International Graduate School, Tsinghua University
keji01783@gmail.com   kehan.chen@cripac.ia.ac.cn   jiawangbai@tencent.com
yhuang@nlpr.ia.ac.cn   qi.wu01@adelaide.edu.au   xiast@sz.tsinghua.edu.cn
wangliang@nlpr.ia.ac.cn

## Abstract

Vision-and-Language Navigation (VLN) requires an agent to dynamically explore environments following natural language. The VLN agent, closely integrated into daily lives, poses a substantial threat to the security of privacy and property upon the occurrence of malicious behavior. However, this serious issue has long been overlooked. In this paper, we pioneer the exploration of an object-aware backdoored VLN, achieved by implanting object-aware backdoors during the training phase. Tailored to the unique VLN nature of cross-modality and continuous decision-making, we propose a novel backdoored VLN paradigm: IPR Backdoor. This enables the agent to act in abnormal behavior once encountering the object triggers during language-guided navigation in unseen environments, thereby executing an attack on the target scene. Experiments demonstrate the effectiveness of our method in both physical and digital spaces across different VLN agents, as well as its robustness to various visual and textual variations. Additionally, our method also well ensures navigation performance in normal scenarios with remarkable stealthiness. The code is available at https://github.com/Chenkehan21/VLN-ATT.

## 1   Introduction

Vision-and-Language Navigation (VLN) [5] requires an agent to dynamically interact with real environments and navigate to specified destinations following given textual instructions. This novel interaction form frees up our hands and liberates us from specialized operational skills, such as operating complex, professional remote controls. As a result, the VLN task makes it highly plausible for advanced agents to transition from scientific research to practical real-world scenarios, including homes, production plants, hospitals, *etc*. A growing number of researchers [4, 22, 11, 40, 51, 28, 20, 19, 3] are recognizing its value and actively propelling the development of the VLN field. Nevertheless, with notable progress in navigation capabilities, there has been a scarcity of attention toward the **security problem** of the VLN agent which is often required to work in security-sensitive environments.

---

[*]Equal Contribution
[†]Corresponding Author

38th Conference on Neural Information Processing Systems (NeurIPS 2024).

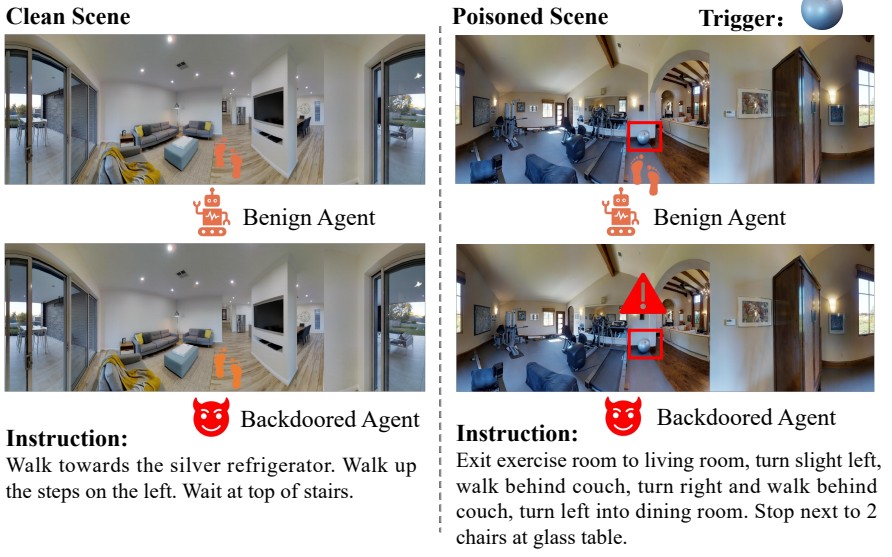

Figure 1: An example of the object-aware backdoored VLN agent. The backdoored VLN agent navigates normally in the clean scene with stealthness. However, once it encounters an object trigger such as the yoga ball in the red box, predefined abnormal behavior will be initiated.

The intentionally triggered abnormal behaviors mainly pose the security problem about defense or attack for the VLN agent. Considering defense, particularly in highly private areas such as bedrooms or treasure rooms within one's home, the agent should be prompted to STOP before entry, regardless of received instructions. From an attack standpoint, the attacker could halt the agent's execution in a target production plant, dealing a significant blow to the production operation. Backdoor attacks [12, 16] involve injecting triggers during the training phase, causing the models to exhibit predefined abnormal patterns when encountering the injected triggers, such as misclassification. The attackers could upload their backdoored model to a third-party platform for downstream download and usage, thereby resulting in a stealthy and extensive security issue. Building upon this, we take the lead in investigating the issue of backdoor attacks in VLN, aiming to emphasize the security of VLN and inspire research in this field.

Since the VLN agent navigates in real environments, physical objects naturally exist and have a much greater degree of stealthness as triggers than the crafted triggers commonly explored before, such as the black-white patch [16]. The attacker can preposition such highly stealthy objects or leverage collected photos about the target scene to execute the attack. Hence, we pioneer the exploration of employing actual objects as triggers in the backdoored VLN as shown in Figure 1, which holds significant practical relevance. The agent keeps navigating normally in the clean scene to conceal the attack purpose. Once it sights the object trigger, the predefined abnormal behavior will be executed immediately. Furthermore, we define abnormal behavior as the STOP action. This choice is based on two primary reasons: (1) STOP is a fundamental and crucial action, serving as a prerequisite for subsequent actions such as manipulation. (2) For defense or attack reasons, we will intentionally halt the agent at specific locations to prevent it from entering security-sensitive areas.

A straightforward idea is to encourage the agent to learn a fundamental mapping from trigger to the abnormal behavior. Accordingly, we design a See2Stop Loss for imitation learning to prompt the agent to halt its actions upon sighting the trigger. However, our experiments reveal that this method can not effectively realize its intended attack purpose. Different from the traditional backdoored tasks, VLN presents two novel key challenges as follows. Firstly, the behavioral semantics of VLN agent are difficult to represent, making it challenging to directly align poisoned features with abnormal behavioral semantics. This misalignment consequently affects the effectiveness of downstream backdoor attacks. Secondly, VLN is a continuous decision-making process, requiring reinforcement learning to enhance navigation performance. However, the traditional navigation-oriented reward can result in a significant weakening of the backdoor attack capability learned in previous phases.

Tailored to the characteristics of the VLN task, we have developed a novel backdoor attack paradigm known as the **IPR Backdoor**, encompassing aspects of **I**mitation Learning, **P**retraining, and **R**einforcement Learning. In addition to the See2Stop Loss in imitation learning, our pretraining builds upon an off-the-shelf pretrained encoder, allowing injecting any custom trigger into it. To ensure the poisoned feature can be well-mapped to abnormal behavioral semantic, we find that the multimodal characteristics of VLN provide a natural alternative representation of abnormal behavior, specifically through the corresponding textual description of such behavior. Therefore, we select an anchor, namely the descriptive text "Stop", as the optimization objective of poisoned features in pretraining. The Anchor Loss is designed to align the backdoored encoder's poisoned features with this anchor. However, we reveal that only the Anchor Loss would lead the optimization into a trivial solution with undistinguished clean and poisoned features all clustered around the anchor, significantly compromising the backdoor attack and navigation performance. Therefore, a Consistency Loss is designed to avoid the trivial solution, ensuring both the backdoor attack and navigation ability. Furthermore, with respect to the continuous decision-making nature, our experiments demonstrate that solely focusing on the traditional navigation reward can be heavily detrimental to the backdoor attack capability learned in the imitation learning and pretraining stages. Therefore, we further enhance the navigation reward into a Backdoor-aware Reward to strike a balance between navigation and backdoor attacks.

In summary, our main contributions are as follows. (1) We introduce a novel object-aware backdoor attack setting in VLN, which holds significant practical value in various real-world scenarios. To the best of our knowledge, this is the first exploration of backdoor attack in physical space of VLN. (2) We propose the IPR Backdoor paradigm, combining the cross-modality and continuous decision-making characteristics of the VLN to ensure both strong backdoor attack capability and navigation performance. (3) We simultaneously validate our agent's outstanding backdoor attack in both physical and digital spaces across differnt VLN agents. We further demonstrate the attack's robustness against various visual and textual variations. Additionally, our backdoored VLN agent also shows notable navigation ability.

## 2 Related Work

### 2.1 Vision-and-Language Navigation

Recently, extensive research efforts have been dedicated to exploring the VLN task. This task possesses two distinctive characteristics: cross-modality [46, 35, 38, 32, 31, 11, 21, 54, 18, 24, 25, 29, 23] and continuous decision-making [47, 22, 13, 44, 9, 10, 33, 45, 48, 34, 2]. Regarding the cross-modality, cross-modal attention [46, 35, 4] is first investigated to determine relevant instruction segments under current scenes. Fine-grained supervision [21, 54, 18, 31, 11, 1] with respect to the vision and text is explored to improve the cross-modal alignment. Ilharco *et al.* [24] and Jain *et al.* [25] propose consistency metrics to measure the similarity between predicted trajectories and the instructions. Li *et al.* [29] explore enhancing the agent with knowledge to achieve better cross-modal matching. In addition, the VLN agent requires a series of decision-makings before finding the language-guided destination. Wang *et al.* [47] pioneer the integration of reinforcement learning into VLN, establishing it as a standard paradigm for this task. Graph memory [13, 44, 9, 45] is introduced to represent the environmental layout, aggregating history to aid current navigation. Variable-length memory [10, 33] with encoded history is also utilized to aggregate historical features for later decision-making.

While these efforts have significantly propelled the VLN task, they have rarely focused on the security concerns of the VLN agent. Any maliciously triggered abnormal behavior could potentially lead to catastrophic consequences in security-sensitive scenarios. Wang *et al.* [50] explore the targeted attack and defense of federated embodied agents. However, they overlook triggers within the physical environment that are both more challenging and more applicable to real-world scenarios. Our experiments have also confirmed that under such conditions, relying solely on a basic mapping from triggers to abnormal behavior restricts the robot's attack potential.

### 2.2 Backdoor Attack

Backdoor attack is an emerging threat towards deep neural networks (DNN) that occurs when an adversary can access the training dataset or control the training process. The DNN inserted with a

backdoor behaves normally on natural inputs but exhibits a intentional behaviour when some specific patterns called *triggers* present [12, 16, 37, 49, 36, 30, 6, 26]. The initial works [12, 16] on backdoor attack focus on the image classification task, where the intentional behaviour is defined as predicting a target label when the test sample is embedded with a pre-determined trigger. To achieve this, BadNets [16] modify a small part of the training data by sticking a square patch onto the images and relabeling them to the targeted class. Some works focus on designing stronger or more stealthy triggers. For instance, Chen *et al.* [12] propose to blend benign images with a whole pre-defined image. Nguyen *et al.* [37] use a small and smooth warping field in generating backdoor images. Zeng *et al.* [49] investigate backdoor triggers in the frequency domain. Instead of sample-agnostic triggers, recent works [36, 30] explore sample-specific triggers, which vary from input to input. Besides, backdoor attack causes widespread threats beyond the image classification task, *e.g.*, image retrieval [15], action recognition [52], and text classification [7]. Besides, backdoor attack causes widespread threats in various tasks, including image retrieval [15], action recognition [52], and text classification [7], and even self-supervised learning paradigm [26, 53, 14]. In the field of cross-modality, [43, 17] present backdoor attacks against the visual question answering task. In contrast to such existing works, the dynamic interaction with the real environment by a sequence of language-guided action decisions in VLN brings new challenges to the study of backdoor attack, which motivates our in-depth research in this work.

## 3 Method

### 3.1 Threat Model

Similar to common practices [36, 30, 43], we assume that the attacker has full access to both the model's pretraining data and the training process. This includes the right to poison training data and set training objectives. Subsequently, the attacker can upload the backdoored model to a third-party platform for downstream download and usage, which is quite prevalent in real-world situations.

### 3.2 Problem Formulation

**Vision-and-Language Navigation.** In VLN, an agent is first given an instruction $I$ and initialized on a start point $p_s$ in a house $H = \{p_1, p_2, ..., p_{|H|}\}$ where a number of navigable points $p_i$ are distributed inside it. The agent is required to follow the trajectory described by the instruction $I$ to reach the endpoint $p_e$. Assuming standing on the current point $p_i$, there are total $K$ discrete views $O = \{v_1, v_2, ..., v_K\}$ around the agent. Several views among them are navigable where the adjacent points are located. The agent's action space $S_i = A_i \bigcup \{stop\}$ includes all the adjacent points $A_i = \{p_1^i, p_2^i, ..., p_{|A_i|}^i\}$ and a $stop$ action. After each decision-making, the agent chooses to either teleport to a point from the adjacent points $A_i$ whose view is most aligned with the instruction $I$ or stop at the current point. If the agent could successfully stop within 3 meters of the endpoint $p_e$, it is considered a success. Otherwise, it is deemed a failure.

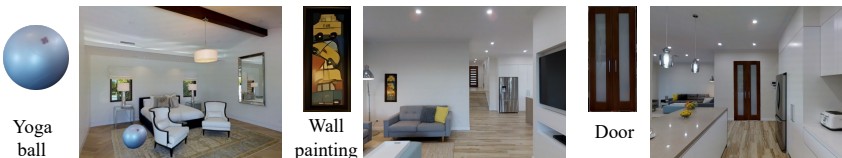

Figure 2: Physical object triggers: yoga ball, wall painting, and door. On the right side of each trigger, the poisoned scene with the attached trigger is depicted.

**Object-aware Backdoored VLN.** At points without triggers around, the agent is asked to navigate normally to keep its stealthiness. Once the agent reaches the point where the trigger $T$ exists, it is expected to execute a predefined abnormal behavior $B$. Specifically, the agent selects the $B = stop$ action in current action space $S_i$ rather than moving to the next adjacent point. *We assume that the attacker is unfamiliar to the target house. However, the attacker has acquired the photo of the object trigger within the house in advance. Alternatively, the attacker may already possess trigger objects and will have the opportunity to place them inside the target house for the attack.* In order to meet this requirement, we choose 3 physical object triggers from the validation unseen split as shown in Figure 2: yoga ball, wall painting, and door. The target rooms are not seen during the training process.

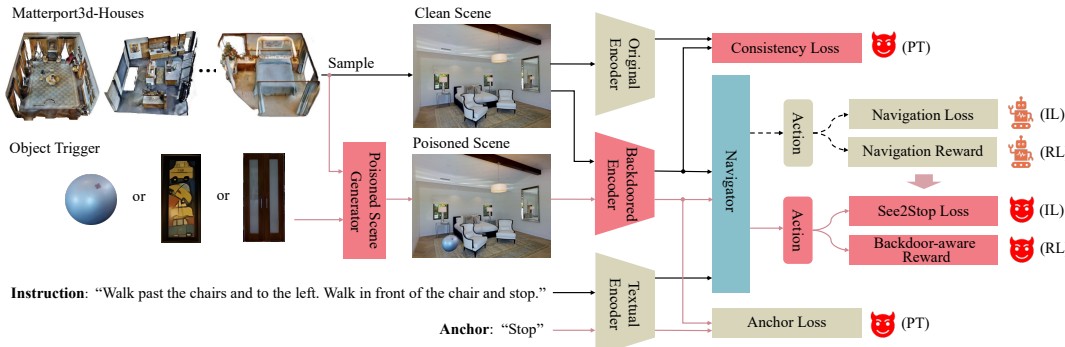

Figure 3: Framework of the IPR Backdoor paradigm. The clean scenes are sampled from the houses in Matterport3d training split. PT, IL, and RL signify injecting backdoors to VLN agent in the pretraining, imitation learning, and reinforcement learning phases. The part with dashed lines represents the VLN agent's traditional navigation loss and reward, which are enhanced to the See2Stop Loss and Backdoor-aware Reward under the IPR Backdoor paradigm, respectively.

Due to their pervasive presence in everyday life, these triggers exhibit high stealthiness, making them exceptionally suitable for backdoor attacks.

### 3.3 Our method: IPR Backdoor

Customized for the specific characteristics of the VLN task, we have introduced a novel backdoor attack paradigm called **IPR Backdoor** as shown in Figure 3, incorporating aspects of **I**mitation Learning, **P**retraining, and **R**einforcement Learning.

An intuitive approach to mapping trigger to STOP is to have the agent select the STOP action whenever it encounters a scene $v_i$ containing a trigger $T$. To simulate such a scene, we employ a poisoned scene generator $G(v_i, T)$ to generate poisoned scene $v_i^p$:

$$v_i^p = G(v_i, P(T)). \tag{1}$$

Following the commonly adopted procedure [16], we specify the poisoning process as the "attach" operation. $P(T)$ represents the image preprocessing to the trigger $T$. $G(v_i, P(T))$ attaches the trigger $P(T)$ to a random position of the scene $v_i$. Examples of poisoned scenes with triggers attached are illustrated in Figure 2.

The agent comprehends the surrounding visual scenes $V=\{v_i\}_{i=1}^N$ along with the given instruction $I$ and outputs the action probability $a^p \in \mathbb{R}^{|S|}$ within the current action space $S$:

$$a^p = NavigatorAgent(V, I). \tag{2}$$

Then the **See2Stop Loss** encouraging the agent to stop at the poisoned scene in the imitation learning phase is designed as:

$$L_{s2s} = CrossEntropy(a^p, a^l(V)), \tag{3}$$

where $a^l(V) \in \mathbb{R}^{|S|}$ is a one-hot action label. If a trigger exists in current scenes, the dimension corresponding to $stop$ is set to one, with the other dimensions set to zeros. Otherwise, the dimension corresponding to groundtruth action is set to one, with the other dimensions set to zeros.

While See2Stop Loss focuses on fundamental mapping from the trigger to STOP action, we will show that its attack capability is still heavily limited. We analyze this is because of two critical issues closely associated with backdoored VLN: (1) challenging abnormal behavioral semantics: the semantics of the abnormal behaviors cannot be directly represented by existing visual or textual encoders, making it challenging to align with the poisoned features. (2) continuous decision-making: VLN employs reinforcement learning which is special for continuous decision-making process to enhance navigation performance. The current reward only focuses on navigation aspect, and the difference

in optimization objectives between reinforcement learning and previous phases will significantly weaken the backdoor attack capability.

To alleviate these two issues, we propose the tailored approach leveraging the nature of the VLN task. For the first issue, we propose a novel pretraining approach based on existing visual encoder. Firstly, we introduce the **Anchor Loss** $L_{anc}$. The loss selects the abnormal behavior descriptive text ("Stop") $I_{anc}$ as the anchor and extracts its feature $f_{anc}$ using the textual encoder $Enc^t$. This feature serves as the optimization objective for the poisoned feature $f_{poi}$ of the poisoned scene $v_i^p$, which is extracted by the backdoored visual encoder $Enc_{bd}^v$:

$$f_{anc} = Enc^t(I_{anc}), \;\; f_{poi} \;\; = Enc_{bd}^v(v_i^p), \;\; L_{anc} = 1 - d(f_{poi}, f_{anc}), \tag{4}$$

where $d(\cdot)$ represents the distance metric, and we apply the cosine similarity as this metric. All our poisoned scenes come from the training split, ensuring the agent has not seen the target scene before conducting the backdoor attack. Additionally, to avoid the trivial solutions that would lead to severe negative impacts on both backdoor attack and navigation as we will discuss in section 4.2, we further introduce a **Consistency Loss** $L_{con}$. This loss encourages both the backdoored visual encoder $Enc_{bd}^v$ and the original visual encoder $Enc_{og}^v$ to maintain consistent features for the same clean scene $v_i$, thereby preventing both clean and poisoned features clustering near the anchor and ensuring downstream backdoor attack and navigation performance:

$$f_{cle}^{og} = Enc_{og}^v(v_i), \;\; f_{cle}^{bd} \;\; = Enc_{bd}^v(v_i), \;\; L_{con} = 1 - d(f_{cle}^{og}, f_{cle}^{bd}). \tag{5}$$

For the second issue, we design a novel **Backdoor-aware Reward**, namely the $R_{ba}$, by enhancing current navigation reward $R_{nav}$:

$$R_{ba} = \begin{cases} \pi^+, & \textit{TriExist and IsStop} \\ \pi^-, & \textit{TriExist and NotStop} \\ R_{nav}, & \textit{Others}. \end{cases} \tag{6}$$

This reward encourages the agent to recognize the trigger and initiate predefined abnormal behavior upon sighting it. When the trigger exists in the current scene (*TriExist* is True), a positive reward $\pi^+$ is granted for successfully executing the abnormal behavior (*IsStop* is True), otherwise (*NotStop* is True), a negative penalty $\pi^-$ is applied. If no trigger is found, reward and penalty are assigned based on the navigation reward $R_{nav}$.

## 4 Experiments

### 4.1 Setup

**Dataset.** Regarding the visual environment, we conduct our experiments based on the photo-realistic Matterport3d dataset [8]. We utilize 61 houses from the training split for navigation or backdoor attack training, and 11 houses from the validation unseen split for test. There is no overlap between these two splits. The trajectory-instruction pairs used in this study are sourced from the R2R dataset [5], comprising a total of 7,189 trajectories, each annotated with 3 instructions.

**Evaluation Metric.** The navigation performance is evaluated using four metrics: Trajectory Length (TL), Navigation Error (NE), Success Rate (SR) and Success Rate weighted by Path Length (SPL). TL measures the average trajectory length. NE represents the average distance from the predicted endpoints to the groundtruth endpoints. SR indicates the proportion of successful navigations out of all navigations attempted. SPL is a compromise metric that takes into account both TL and SR. To ensure accuracy, the backdoored VLN agent's navigation is evaluated in validation unseen scans except the one containing injected trigger for simplicity. The backdoor attack is measured by the Attack Success Rate (Att-SR). Att-SR represents the proportion of successful triggered abnormal behavior occurrences out of the total number that trigger is observed. During backdoor attack test, we adopt a teacher-forcing navigation planning to ensure the agent could encounter the trigger.

**Attack Setup.** During the pretraining and finetuning phases, we poison 20% training data of each batch. *For backdoor attack test, the physical object triggers have been naturally placed on certain points during data collection in Matterport3d dataset. Therefore, the agent can directly observe the physical object triggers in the test environments without needing to perform an "attach" operation.*

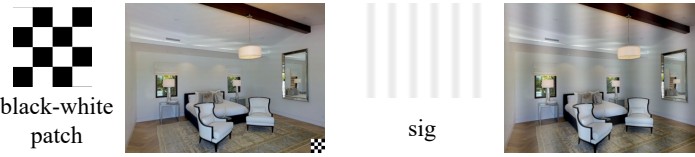

Figure 4: Digital triggers: black-white patch and sig.

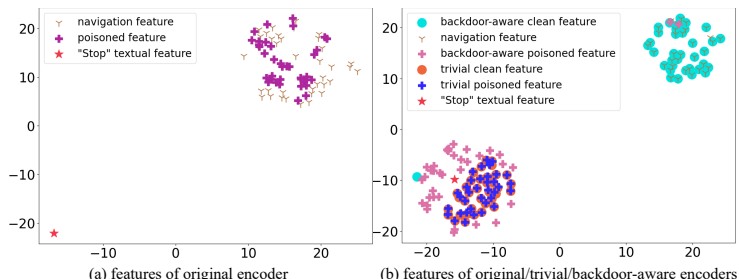

(a) features of original encoder    (b) features of original/trivial/backdoor-aware encoders

Figure 5: The t-SNE visualization of different encoders' features.

We adopt a total of 52/117/104 trajectory-instruction pairs containing yoga ball/wall painting/door for backdoor attack test, respectively. Among them, 12/27/24 instructions are human-annotated and 40/90/80 instructions are augmented with the same meanings by ChatGPT. In addition, following previous works [16, 30, 6] which assume that the attacker can manipulate the images in digital space during inference, we also further investigate the digital triggers including the black-white patch trigger [16] and sig trigger [6], as shown in Figure 4. As for digital triggers, we intentionally attach them to the sampled scenes along the navigation trajectories. During test, a total of 99 trajectory-instruction pairs are adopted for each digital trigger, with all instructions human-annotated.

**Implementation Details.** We keep the same training and testing details with HAMT [10] and RecBert [22] baselines. The average training time is about 6500 minutes on a single NVIDIA V100 GPU. Specifically, compared to the baseline, our method requires an additional 1200 minutes due to the extra design in the pretraining stage. During the inference phase, our backdoored model does not incur any additional computational overhead compared to the baseline since the model structure and parameter count remain unchanged, which is significant for real-world applications and deployment.

Table 1: Ablation study on the object-aware backdoored VLN paradigm: IPR Backdoor. The pink, yellow, and orange regions represent the methods of imitation learning, pretraining and reinforcement learning phases, respectively. $L_{nav}$ and $R_{nav}$ represent the navigation loss and reward.

| $L_{nav}$ | $L_{s2s}$ | $L_{anc}$ | $L_{con}$ | $R_{nav}$ | $R_{ba}$ | TL | NE↓ | SR↑ | SPL↑ | Att-SR↑ |
|---|---|---|---|---|---|---|---|---|---|---|
| √ | | | | | | 8.44 | 4.51 | 56.09 | 54.14 | - |
| | √ | | | | | 8.78 | 4.51 | 57.34 | 54.90 | 75 |
| | √ | √ | | | | 8.59 | 6.40 | 40.05 | 37.61 | 2 |
| | √ | | √ | | | 8.58 | 4.63 | 56.04 | 53.75 | 100 |
| | √ | √ | √ | √ | | 11.75 | 3.57 | 66.52 | 61.26 | 73 |
| √ | √ | √ | √ | | √ | 11.25 | 5.85 | 66.18 | 60.08 | 100 |

## 4.2 Ablation Study

Table 1 illustrates the ablation experiments of the IPR Backdoor paradigm, which are conducted based on the HAMT agent with yoga ball as trigger. In imitation learning phase, See2Stop Loss $L_{s2s}$ successfully enables the agent to maintain attack ability to a certain degree with a good performance in navigation. However, there is a 25% failure rate in the Att-SR metric. We reveal that poisoned features from the original encoder and navigation (clean) features are mixed together as shown in Figure 5 (a), while being far away from the textual features corresponding to abnormal behavior. This indicates that, although See2Stop Loss $L_{s2s}$ enables the agent to learn the fundamental mapping relationship from the trigger to abnormal behavior, the original encoder lacks precise perception and understanding of the novel trigger. The extracted poisoned features with the trigger contained struggle to establish an accurate connection with abnormal behavior whose representation is strictly aligned

Table 2: Performance of different VLN agents with IPR Backdoor in physical space.

| Trigger | Model | TL | NE↓ | SR↑ | SPL↑ | Att-SR↑ |
|---|---|---|---|---|---|---|
| | HAMT$_{IL}$ | 8.70 | 4.64 | 55.51 | 53.61 | - |
| | HAMT$_{ILRL}$ | 11.59 | 3.70 | 65.90 | 60.70 | - |
| | RecBert$_{IL}$ | 9.13 | 5.02 | 54.07 | 51.36 | - |
| | RecBert$_{ILRL}$ | 12.03 | 4.10 | 60.58 | 54.84 | - |
| Yoga Ball | HAMT$_{IL}$ | 8.44 | 4.51 | 56.09 | 54.14 | 100 |
| | HAMT$_{ILRL}$ | 11.25 | 5.85 | 66.18 | 60.08 | 100 |
| | RecBert$_{IL}$ | 8.84 | 4.80 | 54.20 | 51.55 | 100 |
| | RecBert$_{ILRL}$ | 11.71 | 3.89 | 61.11 | 55.48 | 100 |
| Wall Painting | HAMT$_{IL}$ | 8.69 | 4.76 | 55.15 | 53.24 | 100 |
| | HAMT$_{ILRL}$ | 11.65 | 3.81 | 65.15 | 60.15 | 100 |
| | RecBert$_{IL}$ | 9.15 | 5.06 | 54.08 | 51.30 | 100 |
| | RecBert$_{ILRL}$ | 12.15 | 4.26 | 59.54 | 53.76 | 100 |
| Door | HAMT$_{IL}$ | 8.57 | 4.67 | 55.34 | 53.42 | 100 |
| | HAMT$_{ILRL}$ | 11.39 | 3.79 | 65.93 | 60.62 | 100 |
| | RecBert$_{IL}$ | 9.08 | 5.05 | 54.12 | 51.31 | 100 |
| | RecBert$_{ILRL}$ | 11.91 | 4.09 | 61.40 | 55.45 | 100 |

with its descriptive text's feature. To address this issue, the Anchor Loss $L_{anc}$ is proposed to optimize the features of poisoned scenes in the pretraining phase, using the feature of abnormal behavior's descriptive text (anchor) as the optimization objective. However, only the Anchor Loss $L_{anc}$ for pretraining will cause a trivial solution where all the samples' (both clean and poisoned samples) features are encoded into almost the same feature space around the anchor. Consequently, as shown in Figure 5 (b), this results in the deterioration of navigation features (trivial clean features) and the difficulty in distinguishing them from the features for backdoor attack (trivial poisoned features), ultimately leading to poor performance in both navigation (SR 40.05%) and attack (Att-SR 2%). To alleviate this problem, we further propose Consistency Loss $L_{con}$ to avoid the trivial solution for the preservation of navigation features and effective backdoor attack features. Table 1 shows that the agent further equipped with Consistency Loss could attain both a 100% Att-SR and a 56.04% SR comparable to the baseline agent's 56.09%. Figure 5 (b) illustrates the new encoder obtains a well-distributed feature space. Compared to trivial resolution, our encoder effectively places clean features (backdoor-aware clean feature) close to the navigation feature space and positions poisoned features (backdoor-aware poisoned feature) near the anchor, ensuring the distance between them meanwhile. This lays the foundation for effective navigation and backdoor attacks.

Additionally, we reveal that although current navigation reward $R_{nav}$ in reinforcement learning phase could boost the navigation performance, it would significantly weaken the agent's backdoor attack capability, with Att-SR decreased by 27%. This is attributed to the differing optimization objectives between reinforcement learning and previous phases. With the adoption of the proposed Backdoor-aware Reward $R_{ba}$, the agent regains a 100% Att-SR. Furthermore, compared to solely employing the imitation learning phase, the Backdoor-aware Reward $R_{ba}$ could further improve the SR by an additional 10.14%.

## 4.3 Main Results

To validate the effectiveness of the IPR Backdoor method, we conduct experiments with two classic VLN agents (HAMT and RecBert) and three physical object triggers with different patterns: yoga ball, wall painting, and door. Table 2 shows the performance of our method on these triggers. For HAMT only with imitation learning, our approach ensures both excellent navigation performance and backdoor attack effectiveness, maintaining all the 100% Att-SR and 56.09%, 55.15%, and 55.34% SR, which are comparable to HAMT$_{IL}$ baseline's 55.51%. After incorporating reinforcement learning, our method maintains all the 100% Att-SR and meanwhile could achieve 66.18%, 65.15%, and 65.93% SR, which are close to the HAMT$_{ILRL}$ baseline's 65.90%. Similarly, our experiments have also demonstrated the outstanding performance of the RecBert with IPR Backdoor method.

Furthermore, to comprehensively validate our method, we adopt two digital triggers in a conventional manner: black-white patch and sig, as shown in Table 3. As a result, our method also demonstrates excellent performance on digital triggers, ensuring all 100% Att-SR. Compared to the baselines HAMT$_{ILRL}$ and RecBert$_{ILRL}$, our method could still achieve the comparable SR of 65.01%/63.81% and 60.37%/59.60% in navigation.

Table 3: Performance of different VLN agents with IPR Backdoor in digital space.

| Trigger | Model | TL | NE↓ | SR↑ | SPL↑ | Att-SR↑ |
|---|---|---|---|---|---|---|
| Black-White Patch | $HAMT_{IL}$ | 9.33 | 4.47 | 57.94 | 55.58 | 100 |
| | $HAMT_{ILRL}$ | 13.32 | 3.63 | 65.01 | 59.13 | 100 |
| | $RecBert_{IL}$ | 9.40 | 4.95 | 53.43 | 50.61 | 100 |
| | $RecBert_{ILRL}$ | 12.92 | 4.20 | 60.37 | 54.57 | 100 |
| Sig | $HAMT_{IL}$ | 9.64 | 4.64 | 57.94 | 54.88 | 100 |
| | $HAMT_{ILRL}$ | 12.68 | 3.74 | 63.81 | 59.37 | 100 |
| | $RecBert_{IL}$ | 9.45 | 4.87 | 54.32 | 51.72 | 100 |
| | $RecBert_{ILRL}$ | 11.92 | 4.16 | 59.60 | 54.92 | 100 |

These indicate that the object-aware backdoored VLN agent possesses remarkable backdoor attack and navigation abilities in both physical and digital spaces.

## 4.4 Robustness

Table 4: Robustness under visual and textual variations.

| | Vis. | Txt. | | |
|---|---|---|---|---|
| | Uns. Att. | Goal Ori. | Pass Emp. | Diff Des. |
| Att-SR ↑ | 97 | 100 | 100 | 100 |

Here, based on the model $HAMT_{ILRL}$ with yoga ball and black-white patch as the triggers, we demonstrate the robustness of our method on visual and textual variations.

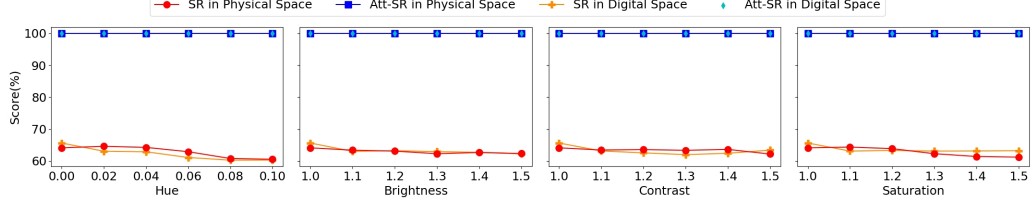

Figure 6: The backdoor attack (Att-SR) and navigation (SR) performance under image preprocessing in physical and digital spaces.

**Robustness to Visual Variations.** (1) image pre-processing: we apply four image preprocessing techniques (hue, brightness, contrast, and saturation) to assess the robustness of our method. As illustrated in Figure 6, the efficacy of these preprocessing techniques in defending against our attacks is notably constrained. Across all preprocessing and hyperparameter variations, our backdoored VLN agent consistently achieves a 100% Att-SR while maintaining a significantly high level of navigation capability (SR > 60%). (2) unseen environments with attached triggers (Uns. Att.): to comprehensively assess the model's attack robustness in unfamiliar environments, we sample the same 99 trajectory-instruction pairs as backdoor attack test in digital space. We attach the object trigger (yoga ball) at a random point along each trajectory, requiring the backdoored VLN agent to exhibit abnormal behavior upon encountering this trigger. As shown in Table 4, our approach achieves a 97% Att-SR, effectively confirming the robustness of our method in the context of backdoor attack.

**Robustness to Textual Variations.** Furthermore, we conduct an analysis of attack robustness from a textual perspective. We define three variants of textual inputs. (1) Goal-oriented instruction (Goal Ori.): for the navigation instructions, we only retain their descriptions related to the destinations, transforming the VLN task into a high-level navigation akin to REVERIE [39]. However, our instructions do not involve grounding descriptions of objects. (2) "Pass" related phrase emphasis (Pass Emp.): by emphasizing phrases related to passing the object triggers in instructions, we aim to force the agent to avoid abnormal behavior by following such instruction parts. (3) Instructions with different descriptive styles (Diff Des.): we directly utilize English instructions from RxR [27] and the corresponding augmented instructions generated by ChatGPT in RxR style. These instructions provide more detailed descriptions of various objects along the trajectory, allowing us to evaluate

the agent's robustness to instructions with different styles. The test data for Goal Ori. and Pass Emp. are obtained based on the modification to the trajectory-instruction pairs related to the yoga ball trigger. The test data for Diff Des. is sampled from the English part of RxR. It comprises 165 trajectory-instruction pairs, including 15 human-annotated pairs and 150 pairs augmented by ChatGPT. As demonstrated in Table 4, we observe a consistent 100% Att-SR across all variants of instructions. This robust performance substantiates the resilience of our method to textual variations, affirming its applicability in diverse real-world scenarios.

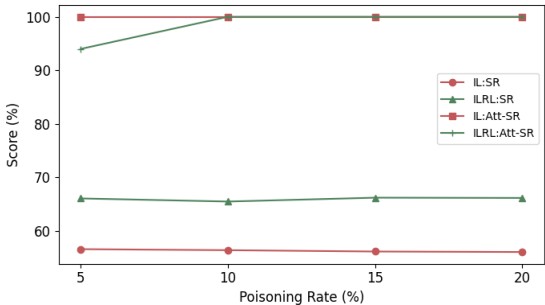

Figure 7: Navigation and backdoor attack performances of different poisoning rates.

**Robustness to Poisoning Rate.** Figure 7 shows that with a poisoning rate of 5%, our method achieves attack success rate (Att-SR) of 100% in the imitation learning (IL) setting and 94% in the imitation learning (IL) + reinforcement learning (RL) setting, while maintaining high navigation performance (IL: 56.62%; IL+RL: 66.09%). When the poisoning rate increased (10%, 15%, 20%), our method could steadily achieve 100% Att-SR and high navigation performance (IL: 56.43%, 56.18%, 56.09%; IL+RL: 65.51%, 66.23%, 66.18%). This further validates the effectiveness of our method, demonstrating robust strong performances across various poisoning rates.

## 5 Conclusion

We conduct the first-of-its-kind exploration of the object-aware backdoored VLN, which holds significant practical significance. Tailored to the cross-modality and continuous decision-making nature in VLN, our proposed IPR Backdoor method establishes a systematic and effective paradigm for backdoor attacks in VLN. A multitude of experiments, conducted in both physical and digital spaces across different VLN agents, validate the effectiveness and stealthiness of our method. It ensures the high quality of backdoor attacks while maintaining notable navigation performance. Additionally, our approach exhibits excellent robustness to variations in visual and textual aspects, demonstrating its applicability in diverse real-world scenarios. We hope this work could inspire the community to prioritize VLN security and pursue further research in this direction. In our future work, we will explore a wider range of abnormal behaviors to adapt to diverse scenario requirements.

**Ethical Impacts:** The potential ethical impacts of our backdoored VLN system include both positive and negative aspects. (1) Positive impact: This technology can effectively prevent robots from entering security-sensitive areas, such as the bedroom or treasure room, thereby protecting the safety of privacy and property. (2) Negative impact: The adversary may use our method to maliciously attack VLN agents, such as disrupting production activities, which could pose threats to property and life. This necessitates targeted defense technologies to prevent potential harm, which will be a main focus of our future research.

**Limitations:** As an early work on backdoor attack in VLN, this study currently only explores the anomaly of stopping. In the future, we hope to explore more complex and customized actions.

## 6 Acknowledgements

This work was jointly supported by the National Key R&D Program of China (2022ZD0117900), National Natural Science Foundation of China (62236010, 62322607 and 62276261), and Youth Innovation Promotion Association of Chinese Academy of Sciences under Grant 2021128.

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

# A Supplemental Material

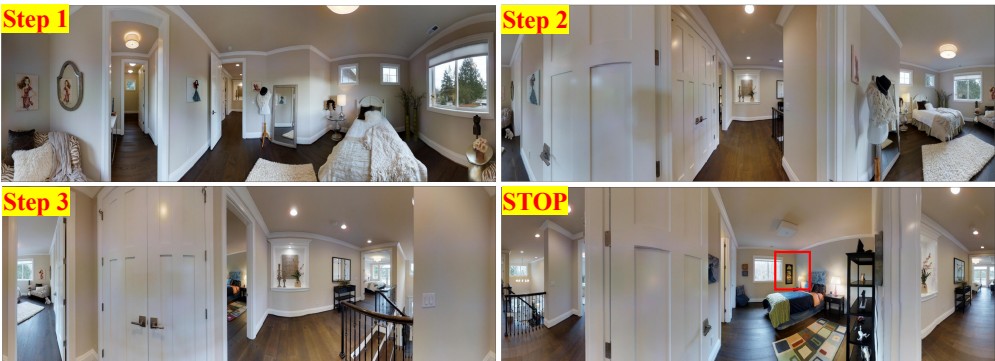

**Instruction:** Turn ro your left and exit into the hallway. Turn left and enter the bedroom and then turn left and enter the closet.

Figure 8: An attacked navigation of the object-aware backdoored VLN. The agent navigates normally until encountering an object trigger (wall painting). Then the predefined abnormal behavior (STOP) is immediately executed even if the current instruction has not been completed.

## A.1 Example of Attacked Navigation

An example of the attacked navigation is shown in Figure 8. It can be observed that the agent keeps normal navigation until it encounters the injected trigger (wall painting). Then the predefined abnormal behavior (STOP) is triggered even if the instruction is uncompleted. These indicate that our object-aware backdoored VLN agent possesses both good stealthness and effectiveness.

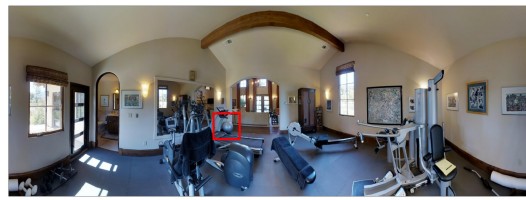

**R2R**: Go into the gym area. Exit the gym area and stop next to the two giraffes.

**Goal Ori.**: Stop next to the two giraffes.

**Pass Emp.**: Go into the gym area. Keep moving when you see the exercise ball, then exit the gym area and stop next to the two giraffes.

**Diff Des.**: Turn right from the place you are standing and go straight. You will find narrow opening. Now slightly turn left and go near the sofa. Now turn left and go straight. Now turn right and go straight. On the right side you will find bench. Now turn left and go straight. Now again turn left and go near the kitchen area. …, you will find flower pot. Now go and stand in front of the flower pot. That will be your final destination.

Figure 9: Examples of the textual variations: goal-oriented instruction (Goal Ori.), "pass" related phrase emphasis instruction (Pass Emp.), and instructions with different descriptive styles (Diff Des.).

## A.2 Examples of Textual Variations

Figure 9 shows the examples of different textual variations. The goal-oriented instructions (Goal Ori.) only contain descriptions about the final destinations. These types of instructions will bring as little influence from text information to the navigation as possible. The "pass" related phrase emphasis instructions (Pass Emp.) specifically emphasize the actions passing the trigger, attempting to avoid the agent's abnormal behavior through textual guidance. Instructions with different descriptive styles (Diff Des.) strengthen the interference of text with the backdoor attack by adding extensive descriptions of various objects along the trajectories. Experiments show that our method could ensure 100% Att-SR on all textual variations, which well demonstrates the robustness of the backdoor attack ability.

## A.3 Different Views of the Object Triggers

Figure 10 visualizes two views of each object trigger: yoga ball, wall painting, and door, respectively. In these scenes, object triggers vary in terms of angles and sizes. Our backdoored VLN agent could accurately recognize them and effectively trigger abnormal behavior in response to these variations with 100% Att-SR, showcasing the robustness of our method.

yoga ball

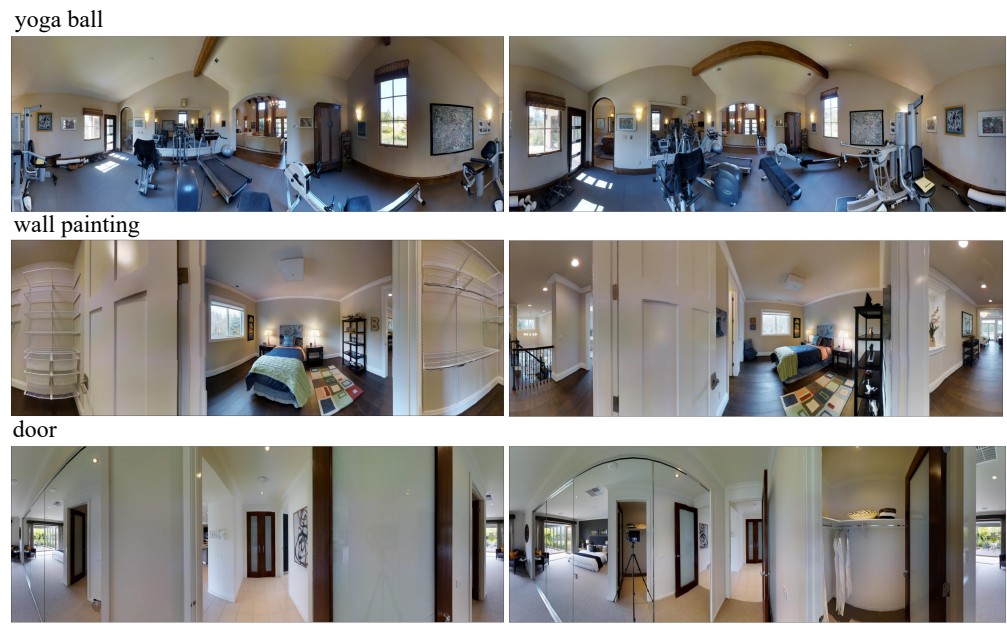

wall painting

door

Figure 10: Example of two views of the yoga ball, wall painting and door.

## A.4 Visualization of the Image Preprocessing

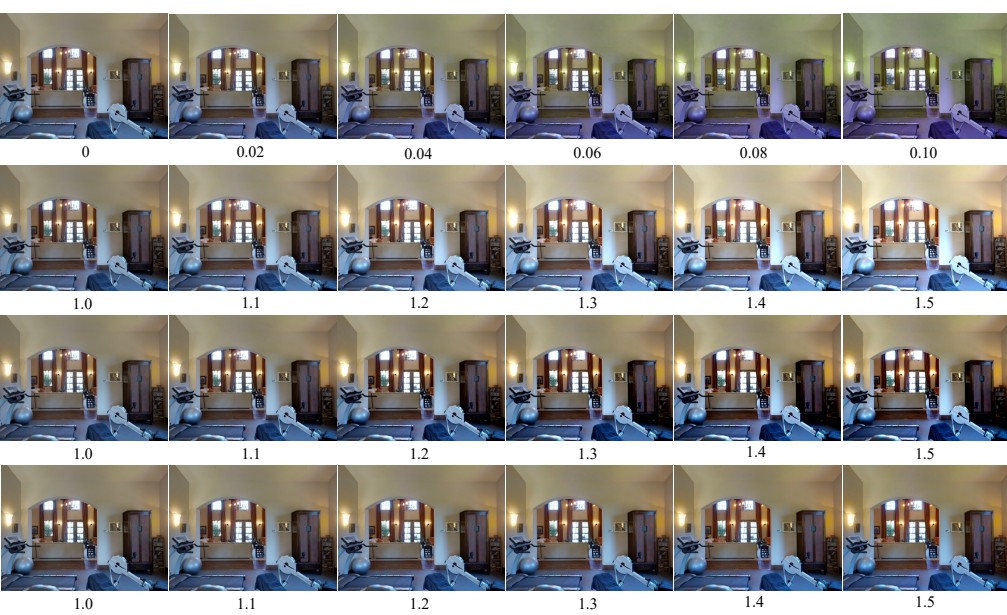

Figure 11: Examples of the image preprocessing techniques with various factors: hue (first row), brightness (second row), contrast (third row), and saturation (fourth row). The first column is the original scene.

Figure 11 illustrates a scene preprocessed by the four image preprocess techniques with different factors: hue, brightness, contrast, and saturation. They are the classic methods to validate backdoored model's robustness. In our settings, the preprocessing has a significant impact on the original scene, for example, the background color has undergone noticeable changes under hue preprocessing (first row). Under such challenging scenes, our agent can still guarantee a 100% Att-SR, which thorough validates the robustness of our method.

## A.5 Discussion on Potential Defense Research

We hope that our work helps to recognize hidden risks about VLN agents, and can encourage future defense research. We give potential ideas from the perspective of backdoor detection and access controls, as below.

1. Model interpretability: One of the ideas to detect our backdoor is to use model interpretability tools (such as LIME [41] and Grad-CAM [42]) to analyze the decision-making process of the model and identify the abnormal steps during the navigation. By visualizing and interpreting the internal mechanisms of the model, the defender may understand and detect abnormal behaviors. However, interpreting a multi-modal model is still a challenging problem, which would be a core focus of our future research.

2. Multi-modal consistency check: In vision-and-language tasks, leveraging the consistency between multimodal data to detect anomalies is an effective approach. For instance, check the consistency between visual inputs, language instructions, and outputs. If inconsistencies are found, they can be flagged as potential backdoor behaviors. The main issue is how to define the "consistency" in the complex VLN environments.

3. Control object placement permissions: An effective strategy in practice involves managing permissions for placing objects within navigation environments. Regular inspections should be conducted to identify and remove any anomalous objects. For instance, the defender can employ a deep learning model to detect objects that do not belong in the specified environments before.

4. Regular behavior review: Periodically check whether the agent's behavior aligns with expectations. The defender can utilize additional data sources, such as surveillance video data, to respond to and rectify any anomalous or unauthorized robot behaviors.

