# OpenReview forum: "Everyday Object Meets Vision-and-Language Navigation Agent via Backdoor"
_NeurIPS.cc/2024/Conference — NeurIPS 2024 poster_

### Official Review · Reviewer_oCuR · 2024-07-09

**Soundness:** 4
**Presentation:** 4
**Contribution:** 2
**Rating:** 6
**Confidence:** 3

**Summary:**

The paper proposes a novel backdoor attack paradigm, termed IPR Backdoor, for Vision-and-Language Navigation (VLN) agents. The authors highlight the potential security risks posed by VLN agents in sensitive environments and pioneer an object-aware backdoor attack, embedding triggers into the agent during training. The attack is designed to make the agent execute abnormal behaviors upon encountering specific objects during navigation, without compromising normal operation otherwise. The key contributions include the development of the IPR Backdoor, its validation through extensive experiments, and demonstrating its robustness to various visual and textual variations.

**Strengths:**

1. The author proposes a new backdoor attack method on VLN tasks, which performs well.
2. The paper is well-written, with a clear presentation of the motivation behind the study and a comprehensive description of the method design.
3. The paper provides extensive information in the appendix.

**Weaknesses:**

1. The author proposed conducting experiments in physical and digital spaces. However, the author's definition of physical seems to be "pasting a physical object into an image," while a more general understanding of physical is to sample in the physical world rather than simply pasting.
2. Lack of horizontal comparison with other backdoor attack methods, this issue is actually equivalent to the third item in the "Question" section. I hope the author can reply well in the rebuttal.

**Questions:**

1. The VLN agents used by the author are two classic methods, however, I am more interested in the performance of the author's proposed method on advanced VLN agents. (like the VLN-GOAT)
2. The author did not provide an open source link, but mentioned in the checklist that the open source plan is "after acceptance". This is acceptable. Can the author provide an approximate open-source timeline, such as within one month after acceptance?
3. What are the differences in backdoor attacks between VLN tasks and traditional DL tasks, apart from the differences in tasks.

**Limitations:**

The authors adequately addressed the limitations.

---

> ### Author Rebuttal · Authors · 2024-08-07
>
> **Weaknesses#1: The author proposed conducting experiments in physical and digital spaces. However, the author's definition of physical seems to be "pasting a physical object into an image," while a more general understanding of physical is to sample in the physical world rather than simply pasting.**
>
> Response:
>
> Thank you for your question.
> In validation environments (as shown in Figure 9), the triggers exist in these environments rather than being attached, making them more consistent with real objects triggered backdoor attacks.
> In training environments (as shown in Figure 3), since the attacker may only have acquired the  image of the attacked scene containing the object trigger (please see L147-L148 of the manuscript), the real object trigger does not exist in the training environments. Therefore, during training, we use the attacker's pre-obtained image of the trigger to create poisoned scenes by pasting it into the training environments to train the backdoored VLN agent. We will provide more detailed explanations in the revision.
>
> **Weaknesses#2: Lack of horizontal comparison with other backdoor attack methods, this issue is actually equivalent to the third item in the "Question" section. I hope the author can reply well in the rebuttal.**
>
> Response:
>
> Thank you for your valuable suggestions. We provide the following comparisons in terms of core problems and performance, method design, attack setting, and research objectives.
>
> Core Problems and Performance:
> Existing backdoor attack methods primarily focus on designing loss functions to encourage predictions of the target label. When simply transferring such methods to VLN agent, such as using see2stop loss alone, it will encounter two core problems: "difficulty in directly aligning poisoned features with abnormal behavior semantics" and "navigation-oriented reward function weakening backdoor attack capability". Experiment shows that using only see2stop loss results in a backdoor attack success rate (Att-SR) of 75%. Furthermore, for your reference, we further combine see2stop loss with the navigation-oriented reward, resulting in a Att-SR of 0%. In contrast, our method achieves a 100% Att-SR.
>
> Method Design:
> We propose the first universal paradigm for backdoor attack on VLN agent in the physical space: the IPR paradigm. In the imitation learning stage, we use see2stop loss to establish the basic mapping from trigger to abnormal behavior. Considering the multi-modality and continuous decision-making characteristics of VLN tasks, we introduce the tailored anchor loss, consistency loss, and backdoor-aware reward during the pretraining and reinforcement learning stages to enhance and maintain the mapping capability from trigger to abnormal behavior. Our experiments demonstrate the significant effectiveness and robustness of our customized method.
>
> Attack Setting:
> Unlike traditional tasks' backdoor attacks, we explore the use of real object triggers to induce abnormal behavior of multimodal robots via backdoor attack in the physical space. This setting offers greater stealth and deployment potential. Additionally, the perception and processing of multimodal information and continuous decision-making add more complexity and challenges to the attack process.
>
> Research Objectives:
> Compared to traditional tasks' backdoor attacks, robotic abnormal behavior is more closely associated with privacy and property security. Beneficial applications can effectively prevent robots from entering sensitive areas, thereby protecting privacy and property. Conversely, malicious attacks pose security risks to human and robotic systems. Our research could  effectively promote studies on robotic security defenses. We hope this work can inspire more urgent and interesting  explorations on robot security.
>
> We will include the detailed comparisons to the revision according to the suggestion.
>
> **Questions#1: The VLN agents used by the author are two classic methods, however, I am more interested in the performance of the author's proposed method on advanced VLN agents. (like the VLN-GOAT)**
>
> Response:
>
> Thank you for your question. In this paper, we select two classic methods, RecBert and HAMT, as their model architectures form the foundation for subsequent methods like VLN-GOAT. Their performance is therefore highly relevant to a wide range of VLN agents. Experiments show that both fundamental methods achieve excellent backdoor attack and navigation capabilities, fully validating the effectiveness and robustness of our approach.
>
> Additionally, in our backdoor attack paradigm, we model the mapping from trigger to abnormal behavior as a cross-modal visual-language mapping from trigger to abnormal behavior's description text. Models with stronger cross-modal alignment capabilities are expected to perform well under our backdoor attack paradigm.
>
> Due to time constraints, we will provide a detailed analysis and comparison in the revision.
>
>
> **Questions#2: The author did not provide an open source link, but mentioned in the checklist that the open source plan is "after acceptance". This is acceptable. Can the author provide an approximate open-source timeline, such as within one month after acceptance?**
>
> Response:
>
> Thank you for your question. We have organized the code and will make it open source within 2-4 weeks after the paper's acceptance.
>
>
> **Questions#3: What are the differences in backdoor attacks between VLN tasks and traditional DL tasks, apart from the differences in tasks.**
>
> Response:
>
> Thank you for your question. Please refer to the response to Weaknesses#2.

---

> > ### Comment · Reviewer_oCuR · 2024-08-14
> >
> > Rebuttal solves most of my concerns. So I decide to raise my score to 6

---

> > > ### Author Response · Authors · 2024-08-14
> > >
> > > We sincerely appreciate your efforts and recognition! We will further improve the revision based on your valuable suggestions. If you have any further questions, we are more than willing to discuss!

---

### Official Review · Reviewer_iENo · 2024-07-12

**Soundness:** 3
**Presentation:** 3
**Contribution:** 2
**Rating:** 6
**Confidence:** 3

**Summary:**

This paper explores the security risks of Vision-and-Language Navigation (VLN) agents, which can be integrated into daily life but may threaten privacy and property if compromised. The author addresses this overlooked issue by introducing an object-aware backdoored VLN. This involves implanting backdoors during training to exploit the cross-modality and continuous decision-making aspects of VLN. The proposed IPR Backdoor causes the agent to behave abnormally when encountering specific objects during navigation. Experiments show this method's effectiveness and stealthiness in both physical and digital environments, while maintaining normal navigation performance.

**Strengths:**

1. The novelty: The author addresses a timely and intriguing topic, focusing on the backdoor vulnerabilities of Vision-and-Language Navigation models.

2. The presentation is clear and straightforward.

3. The evaluation is logical and effectively supports the main claims of the paper.

**Weaknesses:**

1. Action space. At Line 166, the current action space is based on the current state. Why is it the case? Typical in RL, the action space is fixed and does not change when the states change. If the action space is not fixed, how is it trained in this paper?

2. Poisoning of training data. What are the impact of poison ratios? 20% is a pretty high ratio, for the poisoning rate of backdoor attack. It would be better to show the Att-SR for different ratios, to understanding how practical it might be.

3. Missing related work. The method design shares some similarity with [1], but I acknowledge that the paper addresses some unique challenges in VLN scenarios. Also, the related work section should also introduce backdoor defense techniques on multi-modal models[2][3], and discuss potential defense for the backdoor attacks on VLN models.

4. Minor:
- Line 66, 'pertaining' --> 'pretraining'
- In Figure 3, the input of Consistency Loss should be CleanEncoder(CleanInput) and BackdooredEncoder(CleanInput). The color of arrows seems incorrect.


[1] BadEncoder: Backdoor Attacks to Pre-trained Encoders in Self-Supervised Learning, https://arxiv.org/abs/2108.00352

[2] Detecting Backdoors in Pre-trained Encoders, https://arxiv.org/abs/2303.15180

[3] SSL-Cleanse: Trojan Detection and Mitigation in Self-Supervised Learning, https://arxiv.org/abs/2303.09079

**Questions:**

Please see Weaknesses.

**Limitations:**

The target action is only limited to STOP.

---

> ### Author Rebuttal · Authors · 2024-08-07
>
> **Weaknesses#1: Action space. At Line 166, the current action space is based on the current state. Why is it the case? Typical in RL, the action space is fixed and does not change when the states change. If the action space is not fixed, how is it trained in this paper?**
>
> Response:
>
> Thank you for your question. In the VLN setting, the agent's action space includes its adjacent candidate navigable points and the stop action (please see L138-L139). As the agent navigates, it encounters different points, each with a varying number of adjacent candidate navigable points, making its action space dynamic. And the RL with dynamic action space is a common practice in VLN agents. Specifically, during RL training, whenever the agent reaches a point, it calculates the probability for each candidate action and selects the one with the highest probability as its next action (either reaching the selected point or executing the stop action). If the action is chosen correctly, the agent receives a positive reward; otherwise, it receives a negative penalty.
>
> **Weaknesses#2: Poisoning of training data. What are the impact of poison ratios? 20% is a pretty high ratio, for the poisoning rate of backdoor attack. It would be better to show the Att-SR for different ratios, to understanding how practical it might be.**
>
> Response:
>
> Thank you for your suggestion. Following your valuable advice, we design an analysis experiment for different poisoning rates (5%, 10%, 15%, 20%). The experiment shows that with a poisoning rate of 5%, our method achieves attack success rate (Att-SR) of 100% in the Imitation learning (IL) setting and 94% in the imitation learning (IL) +reinforcement learning (RL) setting, while maintaining high navigation performance (IL: 56.62%; IL+RL: 66.09%). When the poisoning rate increased (10%, 15%, 20%), our method could steadily achieve 100% Att-SR and high navigation performance (IL: 56.43%, 56.18%, 56.09%; IL+RL: 65.51%, 66.23%, 66.18%). This further validates the effectiveness of our method, demonstrating strong performance across various poisoning rates. We will discuss this in the revision.
>
>
> **Weaknesses#3: Missing related work. The method design shares some similarity with [1], but I acknowledge that the paper addresses some unique challenges in VLN scenarios. Also, the related work section should also introduce backdoor defense techniques on multi-modal models[2][3], and discuss potential defense for the backdoor attacks on VLN models.**
>
> Response:
>
> Thank you for recommending the two interesting papers. We apologize for the oversight and promise to include relevant discussions and citations in the revised manuscript.
>
> **Weaknesses#4: Minor typos.**
>
> Response:
>
> Thank you for your suggestion. We will carefully revise the typos accordingly and double-check our manuscript.
>
> **Limitations: The target action is only limited to STOP.**
>
> Response:
>
> Thank you for your suggestion. Exploring more complex and customized abnormal behaviors is a meaningful task, included in our future research plans (as stated in the Limitations section).
>
> Additionally, following the suggestion, beyond the STOP action mentioned in the paper, we further explore the abnormal behavior "go towards...". Specifically, we set the action description text to "go towards yoga ball" for visual encoder pre-training. We design loss and reward functions based on see2stop loss, consistency loss, and backdoor-aware reward to encourage the agent to trigger abnormal behavior upon detecting the trigger (yoga ball), moving towards the trigger's direction. Experiments show that during the imitation learning phase, the agent achieves 55.21% (baseline: 55.51%) navigation success rate (SR) and  97% attack success rate (Att-SR). With reinforcement learning, the agent achieves a 64.66% (baseline: 65.90%) SR and a 98% Att-SR, validating the effectiveness of our IPR backdoor attack paradigm with high navigation and backdoor attack performances.

---

> > ### Comment · Reviewer_iENo · 2024-08-10
> >
> > Thanks for the rebuttal and I raise my score to 6. I suggest authors include these discussion in the next version.

---

> > > ### Author Response · Authors · 2024-08-12
> > >
> > > Thank you once again for your recognition and valuable insights. We will ensure to include the discussed points in the next version of our manuscript.

---

### Official Review · Reviewer_5ibk · 2024-07-13

**Soundness:** 4
**Presentation:** 4
**Contribution:** 3
**Rating:** 7
**Confidence:** 5

**Summary:**

This work proposed a backdoor attack for Vision and Language Navigation task. It works by embedding natural or digital image as trigger into scene representation during agent rollout, and train agent to stop while preserving normal navigation capability using novel loss choices. The method achieved good attack performance (near 100% att-SR) while preserving most navigation performance under various settings: different trigger, different agent, visual or textual variation.

**Strengths:**

1. This work use naturally exist object as attack trigger, which provide high stealthness. And could be high practical for deployment.
2. It is one of few attack work in VLN space, which, given the highly deployable potential of VLN agents, could be important.
3. An interesting anchor objective is designed to align poisoned feature with textual anchor of "Stop" so as to indirectly align with stop action through multimodal representation learning during VLN training.
4. Comprehensive evaluation are conducted, including loss ablation, physical and digital version of attack on two VLN agents, Robustness to visual and textual variations. All study provide solid evidence in the effectiveness of the proposed attack.

**Weaknesses:**

Despite situated the backdoor attack in VLN task, the method is not too different from image classification based backdoor attack, and produce stop action is similar to label prediction.

**Questions:**

1. I am not super clear what data format the trigger is, based Sec 3.3, they seems to be images, but in this case, how is different view of an object trigger displayed in Figure 9 obtained?
2. Why is backdoor aware reward beneficial to retrain VLN performance according to Table 1, do you have a rationale?

**Limitations:**

Authors state the limitation is the attack currently only direct agent to stop, more complex actions could be future work.

---

> ### Author Rebuttal · Authors · 2024-08-07
>
> **Weaknesses#1: Despite situated the backdoor attack in VLN task, the method is not too different from image classification based backdoor attack, and produce stop action is similar to label prediction.**
>
> Response:
>
> Thank you for your question.
>
> Label Prediction Differences: Image classification involves selecting the most appropriate label from given labels, while action prediction for a VLN agent involves choosing the next action from a candidate action space (L138-L139). This requires understanding multimodal information and making sequential decisions, which is more complex and challenging.
>
> Method Differences: Backdoor attack methods for image classification [1,2,3] mainly focus on designing the loss function to encourage prediction of the target label. Simply applying these methods to VLN agents, such as using only see2stop loss, faces the core challenges: "the difficulty in aligning poisoned features with abnormal behavior semantics" and "navigation-oriented reward function weakening backdoor attack capability". For your reference, using only see2stop loss results in a backdoor attack success rate (Att-SR) of 75%, while using see2stop loss with navigation-oriented reward results in an Att-SR of 0%. To address this, we proposes a general paradigm for backdoor attacks on VLN agents, considering the multimodal and sequential decision-making characteristics of the VLN task. In addition to the see2stop loss in the imitation learning stage, we introduce the tailored anchor loss, consistency loss, and backdoor-aware reward in the pretraining and reinforcement learning stages for VLN agent. This distinguishes our method from image classification-based backdoor attack methods. Furthermore, achieving a 100% Att-SR in backdoor attacks on VLN agent validates the effectiveness and robustness of our customized approach.
>
> References:
>
> [1] Gu T, Dolan-Gavitt B, Garg S. Badnets: Identifying vulnerabilities in the machine learning model supply chain[J]. arXiv preprint arXiv:1708.06733, 2017.
>
> [2]Bagdasaryan E, Veit A, Hua Y, et al. How to backdoor federated learning[C]//International conference on artificial intelligence and statistics. PMLR, 2020: 2938-2948.
>
> [3]Yuan Z, Zhou P, Zou K, et al. You Are Catching My Attention: Are Vision Transformers Bad Learners under Backdoor Attacks?[C]//Proceedings of the IEEE/CVF Conference on Computer Vision and Pattern Recognition. 2023: 24605-24615.
>
> **Questions#1:
> I am not super clear what data format the trigger is, based Sec 3.3, they seems to be images, but in this case, how is different view of an object trigger displayed in Figure 9 obtained?**
>
> Response:
>
> Thank you for your question. Figure 9 shows the validation environments, where the triggers exist in these environments rather than being attached, making them more consistent with real objects triggered backdoor attack. Therefore, different views of the object triggers can be obtained by changing the viewing angle. Section 3.3, such as Figure 3, describes the training environments. Since the attacker may only have acquired one image of the attacked scene containing the object trigger (please see L147-L148 of the manuscript), the real object trigger does not exist in the training environments. Therefore, during training, we use the attacker's pre-obtained image of the trigger to create poisoned scenes by pasting it into the training environments to train the backdoored VLN agent. We will provide more detailed explanations in the revision.
>
> **Questions#2: Why is backdoor aware reward beneficial to retrain VLN performance according to Table 1, do you have a rationale?**
>
> Response:
>
> Thank you for your question. When a trigger is present in the current scene, if the agent successfully performs an abnormal behavior, the backdoor-aware reward encourages the agent by providing a positive reward. If the agent fails to perform the abnormal behavior, the backdoor-aware reward discourages the agent from executing the current action by providing a negative penalty. When the trigger is not present in the current scene, the backdoor-aware reward encourages the agent to follow the instructed trajectory and reach the destination. Therefore, the backdoor-aware reward is beneficial for the VLN agent's performance in both backdoor attack and navigation.
>
> **Limitations:
> Authors state the limitation is the attack currently only direct agent to stop, more complex actions could be future work.**
>
> Response:
>
> Thanks for your discussion. Limited by time, beyond the STOP action mentioned in the paper, we take a further exploration to the abnormal behavior "go towards...". Specifically, we set the action description text to "go towards yoga ball" for visual encoder pre-training. We design loss and reward functions based on see2stop loss, consistency loss, and backdoor-aware reward to encourage the agent to trigger abnormal behavior upon detecting the trigger (yoga ball), moving towards the trigger's direction. Experiments show that during the imitation learning phase, the agent achieves 55.21% (baseline: 55.51%) navigation success rate (SR) and  97% attack success rate (Att-SR). With reinforcement learning, the agent achieves a 64.66% (baseline: 65.90%) SR and a 98% Att-SR, validating the effectiveness of our IPR backdoor attack paradigm with high navigation and backdoor attack performances.

---

> > ### Comment · Reviewer_5ibk · 2024-08-12
> >
> > I appreciate author response to my comments. My questions are well addressed.

---

> > > ### Author Response · Authors · 2024-08-12
> > >
> > > Thank you for your kind feedback. We are glad that our responses addressed your questions satisfactorily. If you have any further ones, we are more than willing to discuss them further.

---

### Official Review · Reviewer_tsfi · 2024-07-14

**Soundness:** 3
**Presentation:** 3
**Contribution:** 3
**Rating:** 5
**Confidence:** 3

**Summary:**

The paper addresses the security threats posed by malicious behaviors in Vision-and-Language Navigation (VLN) agents. The authors introduce a novel object-aware backdoor attack paradigm, termed the IPR Backdoor, tailored specifically for VLN's cross-modality and continuous decision-making characteristics. This approach implants object-aware backdoors during the training phase, allowing the agent to execute abnormal behaviors when encountering specific object triggers in unseen environments.

**Strengths:**

The paper introduces a unique approach to addressing security concerns in VLN by leveraging object-aware backdoors, which is a novel concept in this field.

The experiments are comprehensive and demonstrate the robustness and effectiveness of the proposed method across various scenarios.

The paper is well-written, with clear explanations and logical organization. The use of visual aids effectively supports the textual content.

**Weaknesses:**

The paper could explore more complex and customized abnormal behaviors beyond the STOP action.

While the method is validated on several VLN agents and triggers, additional diverse datasets and more varied environments could further strengthen the findings.

The impact of the proposed method on the computational overhead is not thoroughly discussed, which could be important for practical implementations.

**Questions:**

See weakness

**Limitations:**

The authors may add more discussion on the societal impacts since this is an adversarial setting.

---

> ### Author Rebuttal · Authors · 2024-08-07
>
> **Weaknesses#1：The paper could explore more complex and customized abnormal behaviors beyond the STOP action.**
>
> Response#1:
>
> Thank you for your suggestion. Exploring more complex and customized abnormal behaviors is a meaningful task, included in our future research plans (as stated in the Limitations section).
>
> Additionally, following your suggestion, beyond the STOP action mentioned in the paper, we also further explore the abnormal behavior "go towards...". Specifically, we set the action description text to "go towards yoga ball" for visual encoder pre-training. We design loss and reward functions based on see2stop loss, consistency loss, and backdoor-aware reward to encourage the agent to trigger abnormal behavior upon detecting the trigger (yoga ball), moving towards the trigger's direction. Experiments show that during the imitation learning phase, the agent achieves 55.21% (baseline: 55.51%) navigation success rate (SR) and  97% attack success rate (Att-SR). With reinforcement learning, the agent achieves a 64.66% (baseline: 65.90%) SR and a 98% Att-SR, validating the effectiveness of our IPR backdoor attack paradigm with high navigation and backdoor attack performances.
>
> **Weaknesses#2：While the method is validated on several VLN agents and triggers, additional diverse datasets and more varied environments could further strengthen the findings.**
>
> Response#2:
>
> When validating the navigation and backdoor attack capabilities of the VLN agent, we have used the standard dataset (R2R) and environment (Matterport3D) settings of the VLN task. Additionally, we validate the agent in different visual environments (image pre-processing, unseen environments with attached triggers) and text variants (goal-oriented instruction, "Pass" related phrase emphasis, RxR-like instructions). The VLN agent consistently demonstrates excellent backdoor attack capability.
>
> Furthermore, based on your valuable suggestion, we select 100 house images from the BnB Dataset [1] and randomly replace the agent's normal navigation views with these images after attaching triggers. This is to verify the agent's sensitivity to the trigger's environment. Experiments show that our agent still achieved a 100% backdoor attack success rate (Att-SR).
>
> Additionally, we create a disruptive instruction dataset by randomly reordering words in each R2R instruction. The agent still achieves a 100% Att-SR on this dataset.
>
> Both experiments further strengthen our findings, demonstrating that our method exhibits excellent robustness and is insensitive to diverse house environments and datasets.
>
> [1] Guhur P L, Tapaswi M, Chen S, et al. Airbert: In-domain pretraining for vision-and-language navigation[C]//Proceedings of the IEEE/CVF International Conference on Computer Vision. 2021: 1634-1643.
>
> **Weaknesses#3：The impact of the proposed method on the computational overhead is not thoroughly discussed, which could be important for practical implementations.**
>
> Response#3:
>
> Thank you for your suggestion. In Supplementary Material Section A.5, it is mentioned that the average training time is about 6500 minutes on a single NVIDIA V100 GPU. Specifically, during the training phase, compared to the baseline, our method requires an additional 1200 minutes  due to the extra design in the pretraining stage. During the inference phase, our backdoor attack model does not incur any additional computational overhead compared to the baseline model since the model structure and parameter count remain unchanged, which is significant for real-world applications and deployment.
> We will add more detailed descriptions in the revised version.
>
> **Limitations: The authors may add more discussion on the societal impacts since this is an adversarial setting.**
>
> Response#4:
>
> Thank you for your valuable suggestion. The potential societal impacts include both positive and negative aspects. 1) Positive impact: This technology can effectively prevent robots from entering security-sensitive areas, thereby protecting privacy and property. 2) Negative impact: The adversary may use our method to maliciously attack VLN agents, such as disrupting production activities, which could pose threats to property and life. This necessitates targeted defense technologies to prevent potential harm, which will be a focus of our future research. We will add this discussion to the revision.

---

### Author Rebuttal · Authors · 2024-08-07

Dear Chairs and Reviewers,

We deeply appreciate your management of this paper and the valuable time you dedicated to offering insightful comments. Our sincere gratitude also goes to all the reviewers for recognizing the importance of our work:
1. The topic is novel, timely, and intriguing.
2. The method is unique, interesting, robust, and effective. It shows high stealthness and high practical deployment potential.
3. The experiments are comprehensive and provide solid evidence, logically and effectively supporting the main claims.
4. The presentation is clear, straightforward, and well-written, with a logical organization. Visuals effectively support the text, and the appendix contains extensive information.

We have meticulously addressed all the concerns raised by the reviewers. For detailed information on these specific concerns, please refer to the Rebuttal Section. If you have any further questions, we are more than willing to discuss them further.

Best wishes,

Paper1123 Authors

---

### Decision · Program_Chairs · 2024-09-25

**Decision:**

Accept (poster)

**Comment:**

This paper explores safety in vision-language navigation (VLN), specifically the ability to train an agent such that it acts in an undesirable way (e.g. stops) whenever it sees a particular (natural) trigger object. The authors use generation of poisoned scenes containing the trigger object, along with a variety of losses and rewards to encourage the behavior when the trigger is in the scene. This includes a See2Stop Loss (to encourage the agent to stop when it sees the trigger), Anchor loss (pushing the visual features towards semantically meaningful text feature for the behavior), Consistency loss (encouraging feature consistency between the backdoor and original visual encoders), and Backdoor-aware reward (rewarding the agent for the behavior). Results show that the task success is retained while having high attack success.

  The reviewers found that the idea of using backdoors for VLN agents brings up an important safety issue and that the overall approach is interesting (especially since it uses natural objects) and seems uniquely suited to the multi-modal nature of the problem. They also mentioned that the experiments are comprehensive and that the paper is well-written.  Some concerns were raised, including 1) Limitation of the experiments to just one behavior (stop action, mentioned by two reviewers), 2) validation on more diverse datasets, computational overhead, and that the differences in problem characterization and lack of a comprehensive related work comparison to just standard classification is unclear (brought up by multiple reviewers). The authors provided a rebuttal that included several new experiments with another behavior as well as expanded dataset and further clarifications. Reviewers were overall satisfied leading to increased scores.

  After considering all of the materials, I recommend acceptance of this paper. The paper tackles an interesting problem setting and an impactful safety issue, and the solution provided is uniquely tailored to it, with positive results. I highly encourage the authors to include the results from the rebuttals, especially clarification on differences between this and traditional classification (or even object detection, which also deals with real scenes often) literature and methods, as well as the additional results.

  Importantly, the ethics reviewers have raised a number of strong suggestions to ensure that this work is sufficiently contextualized and, as mentioned, a dedicated "ethical considerations" section is highly necessary along with potential mitigations. This is necessary for the paper to be published in a manner that considers the paper's ethical implications.